# What Solutions Exist for Developmental Delays Facing Indigenous Children Globally? A Co-Designed Systematic Review

**DOI:** 10.3390/children7120285

**Published:** 2020-12-10

**Authors:** Rona Macniven, Thomas Lee Jeffries, David Meharg, Folau Talbot, Boe Rambaldini, Elaine Edwards, Ian B. Hickie, Margaret Sloan, Kylie Gwynne

**Affiliations:** 1School of Population Health, University of New South Wales, Sydney 2052, Australia; 2Poche Centre for Indigenous Health, Faculty of Health and Medicine, The University of Sydney, Sydney 2006, Australia; tljeffri@gmail.com (T.L.J.J.); david.meharg@sydney.edu.au (D.M.); folau.talbot@sydney.edu.au (F.T.); boe.rambaldini@sydney.edu.au (B.R.); kylie.gwynne@mq.edu.au (K.G.); 3Faculty of Medicine, Health and Human Sciences, Macquarie University, Sydney 2109, Australia; 4Toomelah Public School, Boggabilla 2409, Australia; poche.admin@sydney.edu.au (E.E.); margaret.cobb@det.nsw.edu.au (M.S.); 5Brain and Mind Centre, The University of Sydney, Sydney 2006, Australia; ian.hickie@sydney.edu.au

**Keywords:** child, preschool, language, reward, attention

## Abstract

Early childhood is important for future cognitive and educational outcomes. Programs overcoming barriers to engagement in early education for Indigenous children must address family cultural needs and target developmental delays. This systematic review identifies culturally adapted programs to improve developmental delays among young children, in response to an identified priority of a remote Indigenous community. Five databases (the Cochrane Library, Embase, Medline, Scopus and CINAHL) were searched for English language papers in January 2018. Study quality was assessed, and findings were analysed thematically. Findings were presented to the community at an event with key stakeholders, to determine their inclusion and face validity. Seven relevant studies, published between 1997 and 2013, were identified by the researchers and each study was supported by the community for inclusion. Three studies included on Native American children and four studies included children from non-Indigenous disadvantaged backgrounds. Findings were reported narratively across four themes: storytelling to improve educational outcomes; family involvement improved development; culturally adapted cognitive behavioural therapy to reduce trauma; rewards-based teaching to improve child attention. Limited published research on culturally adapted and safe interventions for children with developmental delays exists but these four themes from seven studies identify useful components to guide the community and early childhood program development.

## 1. Introduction

Educational outcomes are correlated with broad and optimal health and social outcomes. Individuals who complete secondary school are more likely to gain employment, have stable housing, are less likely to have risky behaviours such as substance misuse or tobacco, and as such have improved health throughout the lifespan [1,2]. The countries of Australia, New Zealand, Canada and the United States share similar worldviews that are relational where people and country are holistically connected [3]. These countries also share similar settler-colonial histories where Indigenous peoples were systematically mistreated that has led to an intergenerational impact resulting in the educational, health and broader disparities experienced today [4]. For example, in Australia, Indigenous young people are less likely than non-Indigenous young people to complete secondary education [5] due to a range of factors as wide ranging as access to transport through to entrenched racism. The Closing the Gap strategy is a whole government policy intended to address the disparities in health, education and other key life outcomes of Indigenous Australians. Secondary school completion rates are a priority and whilst year 12 completion rates for Indigenous people have improved over the past decade, significant gaps remain between Indigenous and non-Indigenous students [6]. 

The importance of early childhood education for future cognitive and educational outcomes is well established, particularly for children from low socioeconomic backgrounds [7] who are overrepresented among Indigenous populations [8]. Critical to successful school completion is establishing the foundations of learning including literacy, numeracy and self-regulation, but there are several barriers to establishing these foundations. These include negative past experiences in the education of parents and care givers which have made them distrustful of the school environment and subsequently, reluctant to engage their children in education [9]. The curriculum often does not reflect Indigenous culture and the experience of Indigenous children and families and is therefore less relevant [9]. Furthermore, the cost of purchasing uniforms and other requirements for school may be a barrier for families on low incomes, which is more common among Indigenous families [8]. These barriers combine the impacts in enrolment, attendance, participation and success in education [10]. 

In Australia and internationally, programs have been developed to address the barriers to engagement in early education for Indigenous children and families [11]. These programs have found success when they address the cultural needs of the child and their family in the curriculum. However, an additional barrier to engagement in early education and subsequent educational outcomes is developmental delay. Developmental delay may be apparent in delays in communication and language development [12] as well as behaviour and is more likely to be experienced by Indigenous children [13,14]. It may arise from a number of biological and sociocultural causes, including adverse perinatal outcomes and social deprivation, and be influenced by social determinants of health such as low income and parental education [13]. However, there is limited published literature on how measures or indicators of developmental delay consider Indigenous worldviews [3]. Experience of complex trauma in childhood is also a strong determinant of developmental delay [15]. Trauma-related behaviours are more common in disadvantaged and disengaged communities and for Indigenous people, since the combined effects of colonisation and subsequent harmful government policies such as practices of child removal have contributed to higher population levels of trauma [16]. While trauma is a contributor to developmental delay, the high rates in Indigenous communities are likely to be the consequence of a range of environmental exposures in pregnancy and early childhood. Effective responses to developmental delay include parent-implemented early language and communication programs [12] and early childhood education programs [17].

This review aims to identify culturally adapted interventions to improve developmental delay among young children. This review is undertaken in response to an identified priority of a remote Indigenous community to develop a program to enhance educational outcomes for young children living in this community.

## 2. Materials and Methods

The researchers were approached by a small (<300 population) Indigenous community in central northern New South Wales, Australia, with whom the researchers had a prior relationship to conduct co-designed program delivery and evaluation. The focus of this program was to identify culturally relevant solutions to improve trauma-related developmental delay in Indigenous preschool children. Through a series of community meetings between the researchers and community representatives, the need to undertake a systematic review of international evidence to inform culturally relevant solutions was determined. The co-designed review was guided by the Australian National Health and Medical Research Council Ethical guidelines for research with Aboriginal and Torres Strait Islander Peoples [18]. The co-design process included: discussion with community members about the process and benefits of conducting a systematic review; discussion and agreement about search terms and scope of the review including purpose and intended outcomes; discussion about the included papers; shared thematic analysis focusing on aspects of significance to community members; and agreement about results and conclusions. Co-design discussions were held over a 15-month period between November 2017 and February 2019 face to face at the local school. Resources were provided at each discussion which summarized the work undertaken so far, questions and issues for discussion and proposed next steps.

### 2.1. Protocol and Registration

This systematic review was registered with the International Prospective Register of Systematic Reviews (PROSPERO; Registration no. CRD42018088752) and was conducted and reported following the Preferred Reporting Items for Systematic Reviews and Meta-Analyses (PRISMA) statement Search Strategy [19].

### 2.2. Search Strategy and Inclusion Criteria

Five peer-reviewed relevant online databases were searched: the Cochrane Library, Embase, Medline, Scopus and CINAHL. The search was conducted between January and May 2018. A panel comprising four of the authors gathered to determine the inclusion and exclusion criteria. Search terms to identify global Indigenous groups were used, including Oceanic ancestry group, Indigenous, Aboriginal, First Nation people, Native, Inuit, Sami, Indian, American native, African and European continental ancestry group. Search terms further stratifying the condition or domain being studied were added; neurological and developmental delay, social–emotional development, physiological development, chronic stress, emotional regulation, social competence, hyperarousal, language delay, and stress disorder. We added pre-school aged children and finally terms around intervention to include therapeutics, early intervention, solution, service, program. All types of study design were considered for inclusion. There were no date restrictions, and our search was limited to English language publications. A full search strategy for MEDLINE is provided in Appendix A.

### 2.3. Data Extraction

The panel met to examine the abstracts of all studies that met the inclusion criteria and conduct a risk of bias (quality) assessment. The articles were assessed using the Critical Appraisal Skills Programme (CASP) tool [20] for qualitative research. Data on the study population, type of developmental delay, aim, intervention, findings and main themes were extracted. Data were independently extracted data from each article by one author and were then independently assessed by another author. The studies were then discussed by the panel to determine the final inclusion or exclusion of studies with any doubts resolved by discussion.

### 2.4. Data Synthesis and Interpretation

Upon the completion of the database search and quality assessment, we conducted a content analysis [21] of the findings from the relevant publications to identify common themes. The findings of the potential studies for inclusion were presented to the community at a face-to-face event with key community stakeholders in order to determine their final inclusion and face validity. Following this meeting, the findings were finalized by the researchers. 

## 3. Results

Our search yielded 326 publications (after the exclusion of duplicates). The titles of the 326 publications were reviewed by the primary researcher. A total of 301 publications were initially excluded because they did not fall within the inclusion criteria. The abstracts of 25 publications were reviewed by a panel of four of the authors against inclusion criteria and 10 articles were excluded. The panel separately examined the 15 full-text studies and met together and determined that while none of the studies met each of the inclusion criteria, seven studies were considered to have relevant learnings for the development of the community program. The PRISMA flow diagram for the study is shown in Figure 1.

Two studies focused on Indigenous children [22,23], one study included Indigenous and Hispanic children [24] and for four studies, children were from non-Indigenous disadvantaged backgrounds, specifically Hispanic [25], African American [26], rural Native American [27] and refugee [28] children. These study findings were presented to the community for their views on their relevance and inclusion in the program design and all seven studies were considered relevant for the inclusion by the community and were therefore included in the review. The CASP Qualitative Analysis of the included studies is presented in Appendix A.

Due to the heterogeneity of study designs, a meta-analysis could not be performed, and the findings are reported narratively around key themes. The seven studies were published between 1997 and 2013. Sample sizes varied considerably; five studies took place in the United States [22,23,24,25,26], one took place in the UK [28] and one took place in India [27]. Of the seven studies, two were case studies [22,25], two were cross-sectional or exploratory studies [27,28] and the remaining three were intervention studies [23,24,26]. The citation, study population, type of developmental delay, study aim, design and methodology, intervention, findings and main theme for each study are presented in Table 1.

Four common themes were identified within the seven publications. Firstly, storytelling for children with a traumatic past can be utilized as a tool for literacy improvement. Secondly, a family unit with a father figure showing interest in his child’s education has been shown to improve child development. Thirdly, culturally adapted cognitive behavioural therapy (CBT) has been successfully utilized in children for whom trauma is a barrier to educational development. Finally, a rewards-based teaching method may be an effective way to improve individuals’ attention in class thus contribute to improving educational outcomes.

### 3.1. Theme 1: Storytelling to Improve Educational Outcomes

Four publications identified storytelling as a method to improve educational outcomes in children with developmental delay. Within these, three were focused on the gradual exposure to overcome traumatic experience, two of which were descriptive studies [22,25]. Both proposed narrative play and storytelling as gradual exposure-style therapy as potential solutions to alleviate emotional stress tied to memories of trauma. The third study was an intervention, Story Champs, that used storytelling to improve school preparedness in pre-schoolers [24]. Story Champs is an intervention curriculum that teaches narrative skills to children through focusing on story grammar and complex language, the two levels of narrative language that affect comprehension, when story telling. It offers flexibility for delivery to large or small groups, or to an individual. The use of visual aids such as pictures and retelling personal storytelling practice resulted in experimentally measured improvements to story retells, personal stories, and story comprehension. A further study used storytelling as an intervention for an unspecified learning delay aimed to provide evidence that a better nutritional status leads to better learning outcomes [27]. An additional finding was that children whose fathers read to them led to better learning outcomes. 

### 3.2. Theme 2: Family Involvement Improved Development

The second theme was based on two studies with findings that family involvement improved the emotional and psychosocial development of children during their transition period into school. One study used a school-based mental health service intervention for refugee children that involved a consultation service for staff and students, where teachers referred participants to the service and a mental health worker was assigned to each school [28]. A weekly consultation occurred with the mental health key worker and link teacher, who liaised with other teachers. The study found that refugee children living in the UK who were having difficulties with post-traumatic stress disorder (PTSD) and cultural identification at school showed academic improvement when their parents were willing to take part in finding solutions with the teachers at school. The second descriptive study revealed that young, rural Native American children showed higher levels of psychosocial development if their parents spent leisure time with them, additionally if their fathers took them on outings and if their fathers read to them [27].

### 3.3. Theme 3: Culturally Adapted Cognitive Behavioral Therapy to Reduce Trauma

Two of our studies focused their interventional approach on culturally adapted trauma-focused cognitive behavioural therapy (CBT). CBT is a proven approach to improve traumatic stress symptoms [29]. The Honoring Children, Mending the Circle (HC-MC) program guides a process of therapy through blending Native American traditional teachings with CBT methods, including a relational focus and interconnected spirituality and healing [22]. The Cognitive Behavioural Intervention for Trauma in Schools (CBITS) is a 10-week small group CBT intervention that can be adapted to different cultures and populations [23]. In this review, a quantitative study, aimed at decreasing psychological childhood trauma in sixth graders successfully utilized adapted CBT to improve life equivalent scores (LES) and decrease a child PTSD symptom score among Native American children (CPSS) [23]. The other study utilized culturally adapted trauma-focused CBT through the CBITS to address years of psychological trauma leading to unspecified developmental delay and incorporates elements of Native American spirituality in the treatment [22]. 

### 3.4. Theme 4: Rewards-Based Teaching to Improve Child Attention in an Educational Setting

One study presented a unique solution for classroom behaviour intervention, known as the ‘Good Behavior Game’ (GBG) [30]. GBG is a long established type of interdependent group-oriented contingency management procedure that has been delivered in classrooms and other settings, utilizing team competition and peer influence combined with reinforcement procedures. This study looked at the long-term outcomes of the intervention on students growing up in high-risk areas during their early formative education [26]. The intervention encouraged a positive learning environment by incentivizing good classroom behaviour with peers and teachers. The game was introduced gradually, and rewards were dependent on the overall class behaviour, encouraging a group-based atmosphere. The study presented results 15 years after the students participated and showed lower levels of incarceration and drug-seeking behaviour among participants.

## 4. Discussion

We found seven studies of culturally adapted interventions to improve the cognitive, emotional and developmental delay among young children that were captured within four unique themes. While descriptive studies about developmental delay among priority populations are useful in understanding the issue [31], very few studies propose solutions to the extensive and complex issues surrounding developmental delay. Our review highlights the importance of identifying solutions to such a complex issue and was undertaken in response to an identified priority of a remote Indigenous community to develop an evidence-based program to enhance educational outcomes for children.

While only three studies had included Indigenous children, the findings from each included study were considered by the Australian Indigenous community to be relevant to the community and future program practice, demonstrating active co-design in the review interpretation and finalization. Each of these Indigenous and non-Indigenous population groups experience disparities across many health and educational outcomes [32] and are typically under-represented in research, limiting the evidence base of effective ways to improve health and education equity.

We found that storytelling can improve educational outcomes. The Story Champs storytelling intervention has previously demonstrated clear improvements for children on narrative retell tasks and moderate improvements in personal generation tasks [33]. The present study within this review enhanced these findings with evidence of retell and personal storytelling practice, on story retells, personal stories, and story comprehension [24]. The other study found using superheroes as ‘magical realism’ and a culturally relevant metaphor to overcome trauma [25]. While both studies have small sample sizes which may limit their generalizability, they provide some evidence that storytelling is a culturally relevant method that can benefit children with different forms of developmental delay and may benefit Indigenous children. Further, the oral nature of storytelling, culturally relevant and valued within Indigenous communities [34], is consistent with Indigenous worldviews [3] and storytelling is a common communication method in Indigenous research and practice [35]. This cultural relevance and the use of storytelling over many generations further supports its inclusion in future community program design.

We also found that family involvement improved development. One study also identified story telling as an important part of this family involvement [27]. While this study was descriptive rather than an intervention, the study sample size was large, with over 3500 children, which adds weight to its findings. The association between paternal childcare involvement, especially spending time together, storytelling and taking children for outings and positive psychosocial development. The refugee study [28] found that support from parents at home, as well as mental health service help at school, can improve child adjustment. Therapy treatment models that support traditional beliefs and child-centred parenting practices have been adapted for Native American children through focusing on engagement, language cadence and other culturally relevant factors [36]. An evaluation of the effects of this adaptation would provide important learnings for future Indigenous child program development. The cost effectiveness of a family-centred program for low socioeconomic and ethnically diverse US children that improves academic, behavioural and health outcomes has also been demonstrated [37]. For Indigenous people, family involvement and togetherness is culturally relevant and family-centred early childhood wellbeing interventions in Australia, Canada, New Zealand and the United States have shown benefits for the whole family, including improvements in child emotional and behavioural outcomes as well as cultural revitalization [38].

Our third theme was that culturally adapted CBT can reduce trauma among adolescents. While we did not aim to examine specific therapies in this review, these findings indicate that culturally adapted CBT may be a potential program element. The inclusion of CBT in programs for Indigenous youth is further supported by findings from an adaptation of an established CBT program for Native American adolescents [39] and evaluation [40] that found increases in cultural identity, self-esteem, positive coping strategies, quality of life, and social adjustment. However, it may not be suitable for young children who are the focus of this program development and CBT’s focus on trauma may also mean that is less relevant for overcoming general developmental delays. No known CBT programs for Indigenous Australians currently exist but these learnings from other Indigenous peoples internationally suggest that the approach holds promise yet would likely need further adaptation to local cultural contexts.

Our final theme was that rewards-based teaching can improve child attention in an educational setting. The 15-year follow-up period in this study and its large sample size of over 1000 children gives strength to its findings that a classroom behaviour game, GBG [30], reduces long-term incarceration rates and drug-seeking behaviour [26]. The GBG has been found to be popular, easy-to-use, time-efficient, and widely applicable and versatile [30] and a current trial of an adaptation and enhancement of the GBT in First Nations communities in Canada will give an important future understanding of its impact [41]. Again, we are unaware of the GBG use with Indigenous Australians, but its group-based focus is likely to culturally resonate with children, families and communities [42], with further localized adaptation recommended prior to implementation.

These review findings support the inclusion of each of these four themes in the design and development of the Indigenous community program to improve developmental problems among young children, although CBT may not be relevant for this age group. We also recommend future high-quality intervention trials and the sharing of their findings to inform future policy and practice to best overcome developmental delay among young children, particularly children from priority population groups such as Indigenous people.

The co-design process both enhanced and limited the review. The community owned the review and shaped its focus, approach and analysis. This enhanced the likelihood of identifying literature that would offer culturally relevant solutions. The community preference for a narrow focus that did not included causal factors limited the literature captured in the review. The benefits of the co-design process outweighed the wider search opportunity. The strengths of our study include the identification of practical solutions to inform the development of a community-initiated program to overcome a community-identified issue, developmental delay and subsequent low educational outcomes. Limitations of this review were the small number of participants (≤5) in three of the seven studies and the heterogeneity of study design that did not allow for adequate comparisons across studies. Only English language studies were included due to the review context focusing on English-speaking countries that share similar settle-colonial histories. Nonetheless, the thematic findings provide rich contextual information and were considered by community representatives provide useful learnings to inform the development of a locally designed community intervention.

## 5. Conclusions

There is limited published research focused on culturally sensitive interventions for children with developmental delays. The four themes from seven relevant studies identified in this review provide useful components for a future program for Indigenous young children to improve educational outcomes. The strong co-design element throughout this review means the findings are highly relevant to the Indigenous community and will inform the program design, development and evaluation.

## Figures and Tables

**Figure 1 children-07-00285-f001:**
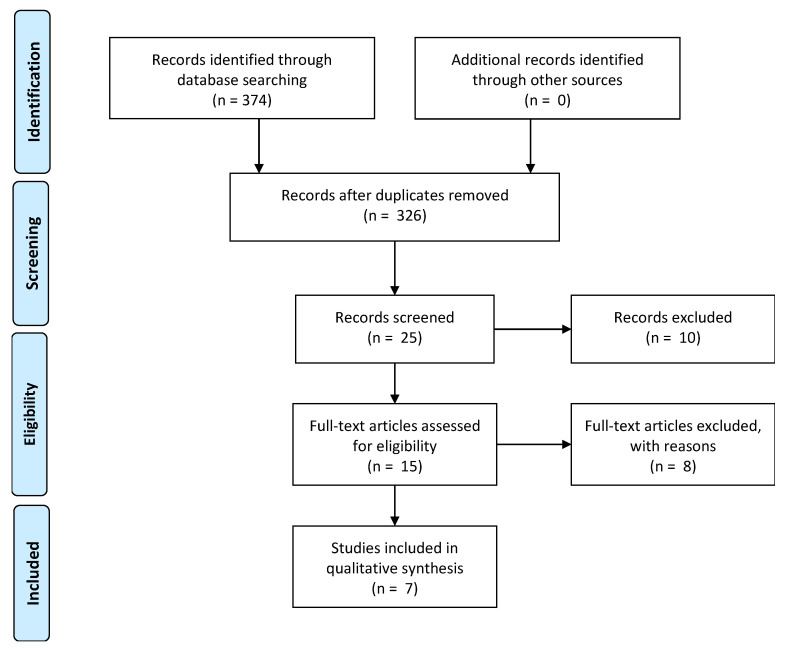
Preferred Reporting Items for Systematic Reviews (PRISMA) flow diagram.

**Table 1 children-07-00285-t001:** Characteristics of the included studies.

**Author, Year Published**	Morsette et al. (2009) [23]	BigFoot et al. (2010) [22]	Kellam et al. (2008) [26]	Vazir et al. (1998) [27]	De Rios (1997) [25]	Fazel et al. (2009) [28]	Spencer et al. (2013) [24]
**Population (*N*; Race, Age)**	48 Native American, 6th grade (11–12 years)	1 Native American, 14 years	1196 urban African American, 1st grade (6–7 years)	3668 rural Native American, 0–6 years	2 Hispanic, 6 and 8 years	47 refugees, 4–19 years	4 Hispanic, 1 Native American (4 years)
**Type of Developmental Delay**	Psychological symptoms following childhood trauma	Trauma exposure related	Behavioural disturbance	Psychosocial and physical delay	Psychosocial reactions to single event trauma	Emotional coping	Developmental disabilities
**Study Aim**	Decrease psychological trauma or post-traumatic stress disorder (PTSD) symptoms	Propose a form of culturally adapted trauma focused therapy	Improve positive classroom outcomes	Assess the psychosocial development of well nourished and malnourished children	Determine whether a novel type of therapy be adapted to a cultural context	Assess intervention effects of for poorly adjusting children	Improve the storytelling ability of learning delayed children
**Design**	Pre-post study	Case study	Cluster randomised controlled study	Cross-sectional study	Case study	Exploratory study	Pre-post study
**Intervention**	Trauma based, culturally adapted cognitive behaviour therapy	Trauma based, culturally adapted cognitive behaviour therapy	Good Behavior Game	Not applicable (N/A)	Storytelling to guide therapy	Mental health service help at school; support from parents at home	Use of storytelling computer software to improve storytelling ability
**Findings**	Improved life equivalent scores and decrease child Post-traumatic stress disorder symptom score scores	The model gives a framework that supports traditional beliefs and children-centred parenting practices	At 15 years follow-up, drug use and incarceration were lower in the intervention group compared to control	Paternal childcare involvement especially, spending time, telling stories, taking child for outings important for positive psychosocial development	Therapy can provide a culturally appropriate intervention to treat the psychological sequelae of trauma	Children receiving direct intervention from mental health services improved in peer problem scoring	There was improved storytelling with Story Champs software
**Summary**	Culturally adapted Cognitive Behavior Therapy (CBT) may ease psychological distress in adolescents	CBT may be a good form of treatment for adolescents who have been exposed to trauma	Good-Behaviour Game may produce long term positive outcomes	Physically and psychosocially well nourished children do better in school compared to those who are deprived	Play therapy, drawing, storytelling may be effective therapy in overcoming trauma in children	Mental health services and family support may benefit poorly adjusted children in the school setting	Story telling can improve the cognition and learning ability of children if adapted prior to formative education

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
