# Peer review of "What Solutions Exist for Developmental Delays Facing Indigenous Children Globally? A Co-Designed Systematic Review"

_children, 2020, doi:10.3390/children7120285_

Round 1

Reviewer 1 Report

Dear authors,

Thank you for the opportunity to review your manuscript. I found the topic interesting and relevant in the field of early childhood and indigenous studies. The paper is sound, the argument is clear, and very well organised. I feel a few aspects would need to be address:

1) Introduction: Since the review is international, the focus on Australia in the introduction is not clearly justified. It would make more sense in terms of the goal of the article to first give a general introduction to indigenous populations in the world, before focusing on the Australian context. Moreover, I feel some considerations from a sociocultural perspective are needed (i.e how much does the notion of developmental delay has actually been constructed from a Western perspective)

2) Co-design: the co-design is mentioned at the beginning of the paper, but not addressed in the methodology and final conclusions. As a reader, I would like to know much more about how the study was co-designed and what were the results of the joint venture, specially: what were the concerns of the indigenous community? what did they want to take out of the research? what was their participation in the design?, among other questions.

3) Limitations: Although you mention the literature review was only carried out in English, I feel the limitations that arise from this should be further analysed. Indigenous studies is a field where publications in other languages are abundant, therefore there might be a significant body of literature that this article is not covering, and maybe extending the search could be suggested for further study.

4) Discussion: the discussion mostly repeats the results. To make the discussion a reacher section I suggest the authors engage in the conversation with the literature they reviewed, in particular, and the field of indigenous studies, in general. 

Reviewer 2 Report

Thank you for working on this critical topic, and more importantly, to respond to people's needs and wants and engaging them into scientific research. It is inspiring to see how participatory research can also be (partly) achieved with literature reviews. Below some remarks and suggestions to further improve your article: 

Introduction: I would be helpful to tighten your line of argument. You are incorporating many concepts in the introduction from secondary school completion, early childhood education, low and socio-economic backgrounds, indigenous vs non-indigenous, trauma, developmental delay, and environmental exposure. Even though they all seem relevant as background information, the objective which follows seems random. Developmental delay is "suddenly" supplemented with cognitive and emotional delay. These are however types of delays. I suggest focusing on one or two types of delays, or, considering your results, stick with the broader term 'developmental delay'.  And no scientific gap on interventions on developmental delay has been put forward. 

Methods: Although you worked systematically in your review, I do not consider this a systematic review. This is a scoping review. Consider revision. Could you include ethical considerations? 

Results: I find this the weakest part of your article. The four themes are relevant and interesting, yet the information about the themes is so meagre. Is there anything more you can share about the interventions? How did these interventions achieve the desired change? Which underlying mechanisms were identified? Which elements did not work? And why? I think that by just naming the interventions and stating they were successful is a missed chance on understanding how and why (not) certain interventions can be applied to other indigenous groups.  

Discussion: Missing critical engagement with the limited amount of studies, and more importantly, the scientific argumentation for using 5 articles from non-indigenous groups. As said, I applaud the co-design, but I would like to see more reflection on how this could have influenced your results. Moreover, based on seven studies, with five from non-indigenous children, I feel you can't draw the conclusions you are drawing now. The work is important as a scoping review, and the first glimpse into interventions, but in my opinion does not yet hold any implications for future practice, rather for future research.  
